# Locating Method for Electrical Tree Degradation in XLPE Cable Insulation Based on Broadband Impedance Spectrum

**DOI:** 10.3390/polym14183785

**Published:** 2022-09-09

**Authors:** Tao Han, Yufei Yao, Qiang Li, Youcong Huang, Zhongnan Zheng, Yu Gao

**Affiliations:** 1National Industry-Education Platform of Energy Storage, Tianjin University, Tianjin 300072, China; 2Electric Power Research of Fujian Power Co., Ltd., Fuzhou 350000, China

**Keywords:** XLPE cable, electrical trees, degradation location, broadband impedance spectrum, distributed parameters

## Abstract

Electrical treeing is one of the main causes of crosslinked polyethylene (XLPE) cable failure. The current methods for locating electrical trees are mainly based on the partial discharge (PD) signal. However, PD signals are easily attenuated in the long cable and the PD test voltage may cause damage to the insulation. This work proposes an improved broadband impedance spectrum (BIS) method to locate electrical trees in XLPE cable. A mathematical model of a long cable containing local electrical tree degradation is established. The Gaussian signal is chosen as the simulated incident signal to reduce the spectral leakage. The location spectrum is obtained by multiplying the frequency domain function of the single-ended reflection coefficient and the Gaussian pulse. It has been found that the location spectrum of the local capacitance change can be characterized as a typical double-peak waveform and the spectrum of the local conductance change can be regarded as a typical single-peak waveform. Electrical tree experiments at different temperatures were carried out to initiate different types of electrical trees. A vector network analyzer (VNA) was used to test the high frequency capacitance characteristics in the treeing process. The location spectra of the 20 m long cable containing different types of electrical trees was calculated by the improved location algorithm. The results show that the location error of local electrical tree degradation is less than 3%. The capacitance of the sliced sample decreases with treeing time. The effect of the bush-pine tree on capacitance parameters is greater than that of the branch-pine tree. A typical double-peak is found in the bush-pine tree location spectrum and a single-peak is found in the branch-pine tree spectrum.

## 1. Introduction

Crosslinked polyethylene (XLPE) cables of 10 and 35 kV are widely used in urban power systems due to their high reliability [1,2,3]. However, insulation degradation is more likely to occur in these cables due to the manufacturing defects and multiple operating environments such as high humidity or temperature [4,5]. Effective methods should be used to monitor and locate the insulation degradation, which is of great significance for the safety of power systems.

Electrical treeing is a typical insulation degradation phenomenon caused by local enhanced electric fields [6,7,8]. When impurities or air gaps exist in the cable insulation, the insulation will be locally broken down where the electric field is concentrated, and then a tree-structure discharge channel will be generated. It is widely reported that electrical trees are one of the main causes for cable insulation failure [9,10].

The current research on diagnosing and locating electrical trees is mostly based on the measurement of the partial discharge (PD) signal. Electrical trees can be located and classified through the time–frequency domain characteristics of PD pulses [11,12,13,14]. Furthermore, some scholars have established multi-dimensional analytical models of PD signals, and intelligent algorithms such as neural networks have been adopted to identify the tree growth process [15,16]. These methods can locate electrical trees in short cable samples. However, PD signals in long cables are difficult to identify due to attenuation and dispersion at high frequencies. At the same time, high voltage is required to generate PD during testing, which may cause secondary damage to the insulation. Therefore, non-destructive location methods for electrical trees in long cables need to be proposed.

One of the non-destructive location methods is time-domain reflectometry (TDR). The TDR method relies on the propagation of an electromagnetic (EM) pulse signal inserted in the long cable. The location of the degradation is identified by analyzing the time domain characteristics of the reflected pulse [17,18]. It was found that water trees and moisture degradation in long cables can be detected accurately by the TDR test [19,20]. Before the TDR test, the pulse width of the EM signal needs to be adjusted so that the bandwidth of the signal can be included in the carrier frequency band of the cable. The carrier frequency of the power cable is related to the geometry of the cable and its length. However, when the information of the tested cable is unknown, the pulse settings need to be tried multiple times according to the operating experience, which increases the difficulty of the TDR test. 

The broadband impedance spectrum method (BIS) has been proposed to avoid the EM pulse signal selecting process. A broadband signal is introduced to the cable and single-ended input impedance of the tested cable is measured in this method. Then the impedance signal is transformed into the location spectrum in the spatial domain by the inverse fast Fourier transform method (IFFT) [21,22]. The bandwidth of the impedance spectrum obtained by the BIS test can cover the whole carrier frequency of the power cable, and so it is not necessary to adjust pulse settings. Based on the above-mentioned measurement principle, some scholars have demonstrated that local heated spots, water tree degradation, and local abrasion of insulation can be located in long cable systems [23,24,25]. The current research on the location algorithm still has the problems of low signal-to-noise ratio and high frequency noise. Moreover, all degradations or defects detected in the above studies have great influence on the dielectric properties of insulation, such as a huge change in insulation resistance or damage to the cable structure. However, electrical trees which occur at a certain point in the cable are different, since they only change the dielectric properties slightly. Therefore, the BIS location algorithm needs to be improved to reduce the interference of high frequency noise and the feasibility of detecting electrical trees in long cables by the BIS method needs to be discussed.

In this work, an improved BIS method was proposed to detect electrical trees in long cables. The transmission line model of long cables containing local electrical trees was established. The location spectra of cables containing capacitance and conductance change were simulated. Then, the effect of electrical tree growth on the transmission characteristics of samples was studied at high frequency in the experiment. Finally, 20 m cables containing different stages of electrical trees were tested, and the experiments verified that BIS can effectively locate electrical trees.

## 2. Experimental Setup

### 2.1. Cable Parameters

In this paper, YJLV-1 × 35 − 8.7/15 kV XLPE cable was used as the tested sample. The cable parameters are shown in Table 1.

### 2.2. Electrical Tree Experiment and Capacitance Measurement

The cable was cut into 1 cm thick slices to study the capacitance change of the cable containing electrical trees. The structure of the sample is shown in Figure 1. The prepared slices and needle electrodes were placed in an oven, where the temperature was controlled at 80 °C for 20 h to eliminate the residual stress. Then, eight needle electrodes were inserted into samples with the same interval, leaving a 2 mm distance between the needle tip and the inner semiconducting layer. The curvature radius of needle tip was 3 μm ± 0.5 μm. Needle electrodes were welded together with wires to prevent the effects of stray capacitance.

Some literature indicates that electrical trees will turn into bush trees from branch trees at high temperatures [6,26]. Different types of electrical trees were also obtained by controlling the temperature of the samples in this work. The tested sample was immersed in silicone oil to prevent flashover and placed in the oven. A voltage of 50 Hz, 10 kV RMS was applied to the needle electrodes. The inner semiconducting layer of the sample and the oven were grounded during the experiment. The images of electrical trees were recorded by a digital microscopic observation system and computer.

The sample was removed from the test platform and cooled for 10 min to room temperature before capacitance measurement. Then, the capacitance of the sample was tested by the impedance analysis module of the vector network analyzer (VNA). VNA model is NA7632A manufactured by Deviser Instruments, Tianjin, China. The needle electrodes were connected to the VNA through alligator clips and the inner semiconducting layer was grounded. The test frequency was set to 50 MHz. The capacitance of the sample was recorded every 10 min in the electrical tree growth. 

### 2.3. BIS Measurement

Since the BIS test is an off-line method, the electrical tree test and the BIS test need to be carried out separately. The tested cable needs to be pretreated before the experiment. The sheathing was removed at both ends of the cable. The 3 cm long wire cores at both ends were reserved and the 5 cm long XLPE at both ends of the cable were reserved to prevent flashover. 

The test platform for the electrical tree in the 20 m long cable is shown in Figure 2a. Needle electrodes were inserted into the cable to generate an electrical tree. The heating tape was tied to the cable near the needle electrode. A voltage of 50 Hz, 10 kV RMS was applied to the needle electrodes to initiate electrical trees and the wire core of the cable was grounded. The applied voltage was maintained every 10 min and then removed. Then, the heating tape was turned off and cooled at room temperature for 30 min. 

The test platform for the BIS test is shown in Figure 2b. A VNA and PC were used to record the data. The cable with electrical trees was connected to the VNA with a crocodile clip and the copper shield was grounded. The test frequency range was set from 100 kHz to 100 MHz with a sampling interval of 10 kHz. 

## 3. Transmission Line Model and Improved Location Algorithm

### 3.1. Transmission Line Model

According to the transmission line theory, when the cable length is greater than one-tenth of the signal wavelength in the BIS test, the distributed parameter model is needed to replace the lumped parameter model. Figure 3 shows the equivalent circuit of the ∆*z*-length transmission line. Distributed parameters *R*, *L*, *G*, and *C* represent resistance, inductance, conductance, and capacitance of unit length, respectively. ∆*z* is the length of the unit transmission line, which is far shorter than the signal wavelength. The cable can be regarded as a series of ∆*z*-length transmission lines.

When the cable is intact, the distributed parameters of each ∆*z* are consistent. The cable is set to be connected to the test port at *z* = 0 and the load is located at *z* = *l*. The impedance of any point in the cable which is located at 0≤z≤l can be obtained through the electromagnetic relationship:(1)$$Z(z)=Z_{c}\frac{1+\Gamma_{V}(z)}{1-\Gamma_{V}(z)}=Z_{c}\frac{1+\Gamma_{V}(l)e^{2jk(z-l)}}{1-\Gamma_{V}(l)e^{2jk(z-l)}}$$
where *Z_c_* is the characteristic impedance of the cable, *k* is the propagation constant which represents the attenuation and dispersion of the traveling signal in the cable, Γ*_V_*(*z*) is the reflection coefficient of *z* which represents the ratio of reflected wave to incident wave at *z*. *Z_c_* and *k* can be calculated by the distributed parameters:(2)$$jk=\sqrt{\left(R+j\omega L)(G+j\omega C\right)}=\alpha +j\beta$$
(3)$$Z_{c}=\sqrt{\frac{R+j\omega L}{G+j\omega C}}$$
where *ω* is the angular frequency of the traveling wave. Since the value of *Z_c_* at *l* is the load, Γ*_V_*(*l*) can be obtained in (1):(4)$$\Gamma v(l)=\frac{Z(l)-Z_{c}}{Z(l)+Z_{c}}=\frac{Z_{L}-Z_{c}}{Z_{L}+Z_{c}}$$
when the end of the cable is open, the impedance of the load is infinite. Bring Equation (4) into (1), and *z* = 0, so the impedance of the test port can be obtained:(5)$$Z(0)=Z_{c}\frac{1+e^{-}{}^{2jkl}}{1-e^{-}{}^{2jkl}}$$

### 3.2. Improved Location Algorithm

In the above discussion, the solution of (5) is only applied to transmission lines in the case of impedance matching. When electrical tree degradation occurs in the cable, the distributed parameters will change at the degradation point. The principle of locating electrical trees by BIS is shown in Figure 4. *Z*(0) can be solved recursively in the cable with multiple impedance mismatch points:(6)$$Z^{\left(n\right)}=Z_{c}{}^{\left(n\right)}(\frac{1+\Gamma_{V}{}^{\left(n+1\right)}\exp[2jk^{\left(n\right)}(z^{\left(n\right)}-z^{\left(n+1\right)})]}{1-\Gamma_{V}{}^{\left(n+1\right)}\exp[2jk^{\left(n\right)}(z^{\left(n\right)}-z^{\left(n+1\right)})]})$$
(7)$$\begin{array}{c} \Gamma_{V}{}^{\left(n+1\right)}=\frac{Z^{\left(n+1\right)}-Z_{c}{}^{\left(n\right)}}{Z^{\left(n+1\right)}+Z_{c}{}^{\left(n\right)}}(Z^{\left(m\right)}=Z_{L},n=0,1,2,\ldots ) \end{array}$$

By (6) and (7), it can be found that *Z*(0) is only related to the distributed parameters of the cable itself. In the BIS test, *Z*(0) is equivalent to BIS, which can be measured directly by the VNA. Therefore, BIS can be regarded as an analysis tool for the change of the cable insulation condition. The IFFT method has been used to convert the real or imaginary part of BIS into a time-dependent signal and degradation can be located by identifying where the amplitude increases [17]. To improve the resolution of the test, the reflection coefficient method is used in this paper. The reflection coefficient of cable at the beginning of the cable can be obtained by BIS:(8)$$\Gamma_{v}(0)=\frac{Z(0)-Z_{i}}{Z(0)+Z_{i}}$$
where *Z_i_* is the wave impedance of intact cable, and it can be obtained by the parameters of the cable:(9)$$Z_{i}=\sqrt{\frac{\mu_{0}}{\varepsilon_{0}\varepsilon_{r}}}\frac{\ln(r_{s}/r_{c})}{2\pi }$$
where *μ*_0_ is the permeability of a vacuum, *ε*_0_ is the dielectric constant of a vacuum. Since Γ*_V_*(0) reflects the ratio of reflected signal and incident signal at the beginning of the cable, the frequency-dependent reflected signal can be obtained by multiplying the frequency-dependent incident signal by Γ*_V_*(0). Then, the time-dependent signal can be obtained through IFFT. In this work, the Gaussian pulse signal is selected as the incident signal during IFFT. The time–frequency domain conversion process is shown as follows:(10)$$S_{i}(t)=ae^{-\frac{{\left(t-b\right)}^{2}}{2c^{2}}}$$
(11)$$S_{i}(w)=\sqrt{2\pi }acf_{Gaussian}e^{-(\frac{w^{2}}{2{\left(\frac{1}{2\pi c}\right)}^{2}})}$$
(12)$$s_{r}(t)=\mathrm{abs}[\mathrm{IFFT}(S_{i}(w)\Gamma_{v}{\left(0\right)}^{*})]$$
where *a* is the maximum amplitude of the Gaussian signal, *b* is the time shift of the signal, and *c* is related to the pulse width. *S_i_*(*t*), *S_i_*(*w*) are the time domain function and frequency domain function of the Gaussian signal. *f_Gaussian_* is the sampling frequency of the Gaussian pulse. Γ*_V_*(0)^*^ is the frequency domain continuation of Γ*_V_*(0), and its value in the negative frequency domain is the conjugate of that in the related positive frequency domain. *S_r_*(*t*) is the time domain function of the reflected signal. The amplitude of *S_r_*(*t*) is used as the location spectrum of the cable. The pulse width is the main factor influencing the frequency domain characteristics of the incident signal. Full width at half maximum (*FWHM*) of *S_i_*(*t*) can be calculated by:(13)$$FWHM=2\sqrt{2\ln2}c$$

The traveling time of the swept frequency signal in the intact cable can be calculated from the wave speed of the electromagnetic wave in (14). The electrical trees have a slight impact on the overall dielectric properties of the cable, and so the traveling time in the cable containing electrical trees can be approximated by the results in (14).
(14)$$T=\frac{2l}{v}=2\sqrt{\mu_{0}\varepsilon_{0}\varepsilon_{r}}l$$

Since *a* and *b* have no effect on the amplitude–frequency characteristics of the pulse, *a* = 1, *b* = 1 × 10^−6^ are fixed in this work. The value of *FWHM* is determined by *c*. The amplitude–frequency diagram of the Gaussian pulse with different *FWHM* are shown in Figure 5. It can be found that the result after FFT of *S_i_*(*t*) is still a Gaussian function. *S_i_*(*w*) is a function that has the largest amplitude at 0 Hz and attenuates as frequency increases. In this work, the upper limiting frequency of Γ*_V_*(0) is 100 MHz. In order to avoid the spectrum leakage at high frequency, the Gaussian signal should be attenuated to zero at 100 MHz. When *FWHM* = 0.05 *T*, the amplitude at 100 MHz is 1.138 and the amplitude at 0 Hz is 43.005. When *FWHM* increases, the amplitude at 0 Hz increases and the amplitude at 100 MHz decreases. At the same time, when the *FWHM* increases, the width of the reflected peak will increase, which will affect the resolution of the location spectrum. According to the principle of choosing a smaller *FWHM* under the premise that spectrum leakage does not occur, the Gaussian pulse with *FWHM* = 0.075 *T* is selected in this paper.

### 3.3. Simulation of Location Algorithm

From the above discussion, it can be found that the results of BIS are only related to the change of cable distributed parameters, including *R*, *L*, *G*, and *C*. The distributed parameters of coaxial cables can be calculated by (15)–(18):(15)$$R(w)=\sqrt{\frac{\mu_{0}w}{8\pi^{2}}}(\frac{1}{r_{c}\sqrt{\gamma_{c}}}+\frac{1}{r_{s}\sqrt{\gamma_{s}}})$$
(16)$$L(w)=\sqrt{\frac{\mu_{0}}{8\pi^{2}w}}(\frac{1}{r_{c}\sqrt{\gamma_{c}}}+\frac{1}{r_{s}\sqrt{\gamma_{s}}})+\frac{\mu_{0}}{2\pi }\ln\frac{r_{s}}{r_{c}}$$
(17)$$G=\frac{2\pi \gamma_{r}}{ln\left(r_{s}/r_{c}\right)}$$
(18)$$C=\frac{2\pi \varepsilon_{0}\varepsilon_{r}}{ln(r_{s}/r_{c})}$$

*R* and *L* are related to the properties of the conductors in the cable. *G* and *C* are determined by the electromagnetic properties of the insulation. Since the degradation mainly affects the properties of insulation, the influence of degradation on *G* and *C* is discussed here.

A group of cables are set to simulate the change trend of different *G* and *C* when local degradation occurs. The characteristics of different degradation in simulated cables are set in Table 2. *G*_0_ and *C*_0_ in Table 2 are the distributed parameters of the intact cable which can be calculated by (17) and (18). In the settings of simulation parameters in Table 2, the reason why the change of the conductance is larger, and the change of the capacitance is smaller, is that the BIS is more sensitive to the change of capacitance.

Figure 6 shows the location spectra of different simulated cables. It can be found that when the capacitance changes locally in the cable, there are two reflected peaks at the degradation location. In contrast, there is only one reflected peak at the degradation location when the conductance changes in the cable. Here, 1# and 2# simulate cables with local reduction of capacitance. The location spectra of 1# and 2# show the same changing trend. The amplitude of the first reflected peak in the location spectra are larger than that of the second one. At the same time, the peaks of two reflected peaks of 2# are larger than those of 1# cable because a bigger capacitance change exists in 2#. In this study, 3# and 4# simulate cables with a local increase of capacitance. The peak of the first reflected peak is smaller than that of the second one in both location spectra of 3# and 4#. The amplitude of reflected peaks in 4# with a bigger capacitance change is higher than that of 3#. Here, 5#, 6#, and 7# simulate cables with a local increase of conductance. The location spectrum of 5# changes slightly, while the location spectrum of 6# cable has a reflected peak with a small amplitude at the degradation location and the reflected peak in 7# has a larger amplitude. It should be noted that when the distributed parameters change slightly, the reflected peak value at the change point may be smaller than the amplitude without change (1#, 3#, 5#, and 6#). When the amplitude of the reflected peak is too small, the peak may be confused with the noise signal, which affects the identification of the reflected signal.

According to the simulation, it can be found that the location spectrum is more sensitive to the change of capacitance than conductance. There will be a reflected peak in the location spectrum only when the conductance was changed by more than 13 magnitude orders. 

Table 3 shows the locating results of different simulated cables. It is more accurate to locate capacitance change by the midpoint of two reflected peaks (1#, 2#, 3#, and 4#). On the other hand, when locating conductance change, the location of the fault can be accurately judged by the location of the peak with maximum amplitude (5#, 6#, and 7#). Locating capacitance change with the largest peak location can cause large errors in the locating results. When the capacitance decreases, the location of largest peak shifts to the measurement port (1#, 2#), and when the capacitance increases, the location of the largest peak shifts to the load port (3#, 4#).

The cable containing electrical trees is not completely broken down, and so the conductance will not change by orders of magnitude. Therefore, the capacitance is the main factor affecting the location of the cable containing the electrical trees. The location spectra of cables containing electrical trees and the influence of electrical trees on the capacitance parameters will be studied by experimental methods below.

## 4. Results and Discussion

### 4.1. Electrical Trees

Figure 7 shows an electrical tree grown in the sliced sample at 30 °C. The shape of the tree at 30 °C can be characterized as a branch-pine tree. The electrical tree grows slowly for 30 min after initiation (Figure 7a–c), and then the pine structure develops from the main channel outwards (Figure 7d–f). The growth rate of pine structure is faster, and its color is lighter. The branch-pine tree grows to breakdown after 90 min. The growth process of the electrical tree in the sample at 70 °C is shown in Figure 8. The shape can be characterized as representative of a bush-pine tree. The tree structure is denser and in a deeper color. The bush-pine tree grows slowly for 60 min after initiation (Figure 8a–c) and then the tree develops to breakdown after 150 min. The growth rate of trees grown at 70 °C is slower than those grown at 30 °C. 

### 4.2. Capacitance Results

The degradation area changes with the growth of the electrical tree. Accumulative damage (AD) is defined as the damaged area caused by the electrical tree. AD is calculated by summing the number of pixels in the tree area. Figure 9 shows the relationship between the sample capacitance at 50 MHz and the AD of the electrical tree. The linear fitting of the capacitance value and the AD value shows that the capacitance decreases as the electrical tree grows. With the branch-pine tree, the capacitance is 1.418 pF before initiation, and the capacitance becomes 1.368 pF at 90 min. For the sample with the bush-pine tree, the capacitance is 1.405 pF before initiation, and the capacitance is 1.314 pF when the electrical tree grows for 150 min. It can be found that the AD of the sample with the bush-pine tree increases faster as the trees grow, and the reduction in capacitance is also larger. 

Figure 10 shows the model of distributed parameters during the growth of electrical trees. The electrical tree can be regarded as parallel to *n* channels [27]. *C_n_*′ represents the capacitance between needle tip and the *n*th channel end. *C_n_* is the capacitance between the channel end and the ground. The channels of the electrical tree can be divided into two types: conductive and non-conductive. There are a lot of conductive carbonized materials existing in the conductive channels. On the contrary, there are none or only a small amount of conductive materials inside in the non-conductive channels [28]. Therefore, when the non-conductive channel grows, the composition of the channel is mainly the air gap with a smaller relative dielectric constant. When the non-conductive tree grows, *C_n_*′ will continuously decrease. The capacitance between the needle tip and the ground electrode can be regarded as a series of capacitances. When *C_n_*′ decreases, the overall capacitance of the sample will also decrease. The experimental result in Figure 9 is consistent with this. 

### 4.3. Location Results

The location spectrum of the tested cable is shown in Figure 11, which is obtained by time–frequency transformation of the measured BIS. The Gaussian pulse signal with *FWHM* = 0.075 T is used in time–frequency domain transformation. The cable containing the electrical tree is not broken down, so the conductance does not change by orders of magnitude according to the simulation. Two consecutive reflected peaks appear at the measurement port caused by impedance mismatch between alligator clips and cables. A larger peak appears at the load port of the cable. The load port remains open circuited during the test, so the traveling wave is totally reflected at the load location. Since the amplitude of the reflected peaks caused by the degradation is much smaller than the reflected peaks at both ends of the cable, a blind zone exists in the BIS test. When *FWHM* of the selected Gaussian incident signal increases, the area of the blind zone will increase.

Figure 12 shows the location spectrum of the 20 m cable containing a branch-pine tree at different growth times. The location of the electrical tree is about 8.2 m away from the cable measurement port. It can be seen that when the electrical tree grows for 20 min, the reflected peak is not obvious, and the location errors may occur due to signal interference. As the growth time of the electrical tree increases, the amplitude of the reflected peak increases. When the growth time is 120 min, the amplitude of the reflected peak reaches 1.77 × 10^−3^. 

As in the above discussion, the change in capacitance is the main factor for reflection at the degradation location. Therefore, two consecutive reflected peaks should appear in the location spectrum. However, only one reflected peak appears in each location spectrum in Figure 12. The reason for the reduction in the number of reflected peaks is that the branch-pine tree has little effect on the capacitance. It can be seen from the simulation in Figure 6 that when the capacitance change is small, there is a large difference between two reflected peaks. The larger reflected peak is easily identified, and the smaller one is easily confused with other noises.

Figure 13 shows the location spectrum of the 20 m cable containing a bush-pine tree at different growth times. The location of the electrical tree is 9.3 m. With the increasing tree growth time, two reflected peaks with large amplitude appear in the location spectrum. When the electrical tree grows for 20 min, the peak values are 7.68 × 10^−3^ and 3.04 × 10^−3^, respectively. Then, the peak values rise to 1.07 × 10^−2^ and 6.04 × 10^−3^ when the tree grows for 60 min. When the electrical tree grows for 120 min, the peak values reach 1.28 × 10^−2^ and 7.64 × 10^−3^.

When the bush-pine tree grows at a certain point in the cable, the capacitance of this point decreases obviously according to the results above. Therefore, there should be two reflected peaks at this point, and the peak of the first peak is larger than the second peak. The experimental result in Figure 13 is consistent with this.

### 4.4. Analysis of Location Error

When the local capacitance changes, the degradation should be located by the midpoint of the two reflected peaks. However, the branch-pine trees have little influence on the capacitance, so the second reflected peak value is too small to be recognized in the growth stage of electrical trees. Therefore, the branch-pine tree can only be located by identifying one reflected peak and the location is judged by the location of this peak. On the contrary, there are two obvious reflected peaks in the spectrum of the bush-pine tree and the location can be judged by the midpoint of the two reflected peaks.

The location error can be calculated by (19):(19)$$Error=\frac{l_{s}-l_{m}}{l}$$
where *l_s_* is the degradation location in the location spectrum, *l_m_* is the measured actual degradation location, *l* is the length of the tested cable. The location error of the location spectrum for different types of electrical trees is shown in Figure 14. It can be found that the locating error of locating the branch-pine tree is larger and the error is negative during the growth of the electrical tree.

The reason for the error is that there is an unrecognized reflected peak in the spectrum. When the branch-pine tree develops, the first reflected peak is selected to be the key location factor, so the location is closer to the measurement port. The smaller error in locating the bush-pine tree proves that it is more accurate to locate capacitance change through the midpoint of peaks.

In further work, we will search for new incident signals which show higher positioning accuracy and resolution. The waveform and the amplitude–frequency characteristics are our main considerations for new signals. Meanwhile, the denoising method for BIS will be the main subject in our further research.

## 5. Conclusions

In this work, an improved BIS algorithm was proposed. The location sensitivity was improved by changing the pulse width of the incident signal. The effect of conductance and capacitance on the location spectrum was analyzed. The location spectrum of long cables with different stages of electrical tree degradation was measured. The reflected peak waveform of the impedance spectrum was explained by analyzing the influence of the electrical tree on the distributed parameters. The main conclusions are as follows:A Gaussian pulse with *FWHM* of 0.075 T was selected as the simulated incident signal. The location spectrum obtained by multiplying the reflection coefficient and the incident signal shows an improved locating accuracy and reduced spectral leakage.The location spectrum shows double peaks at changed capacitance and a single peak at changed conductance. The midpoint between two peaks is used to locate capacitance change, and the location of the maximum reflected peak is used to locate conductance change.The capacitance will decrease with the growth of electrical trees. The influence of the bush-pine tree on capacitance was greater than that of the branch-pine tree. The location spectrum shows double reflected peaks when bush-pine trees grow and shows one single peak when branch-pine trees grow.By improving the BIS algorithm, the location error of the branch-pine tree was less than 3% and the location error of the bush-pine tree was less than 1%.

## Figures and Tables

**Figure 1 polymers-14-03785-f001:**
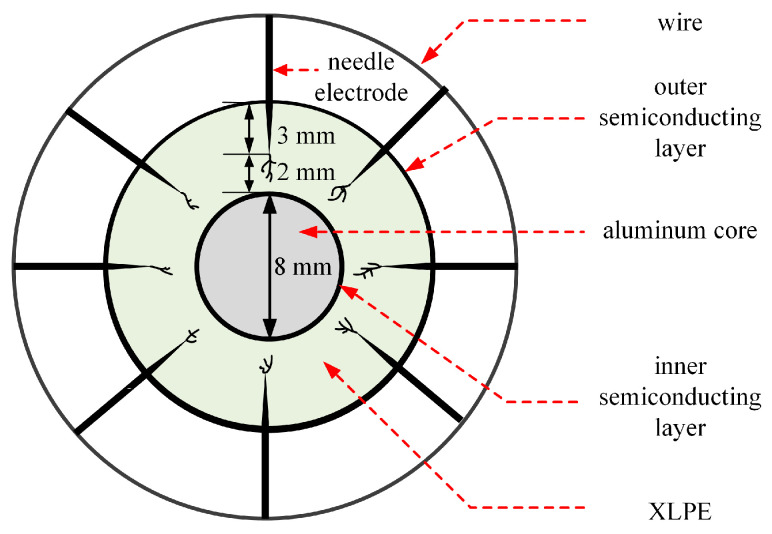
The structure of the sliced cable sample.

**Figure 2 polymers-14-03785-f002:**
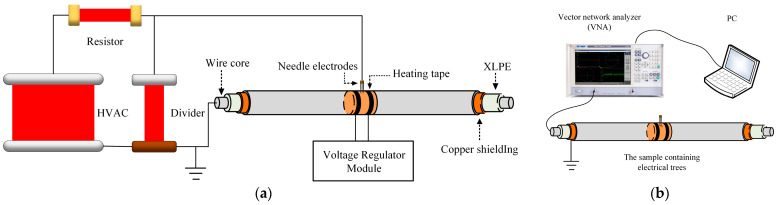
BIS test system of cables containing electrical trees: (**a**) AC electrical tree test circuit diagram; (**b**) BIS circuit diagram.

**Figure 3 polymers-14-03785-f003:**
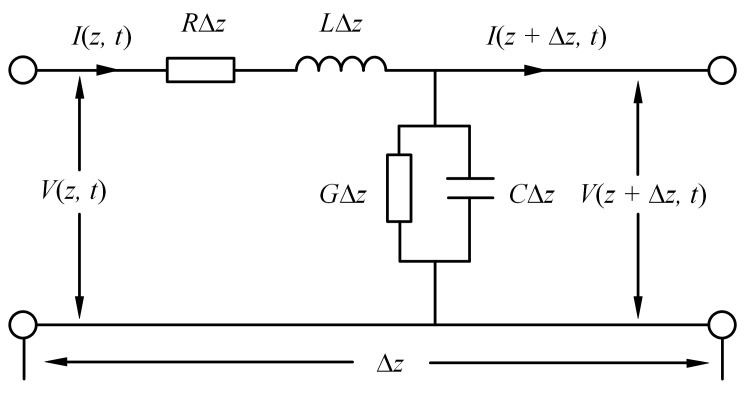
∆*z*-length transmission line.

**Figure 4 polymers-14-03785-f004:**
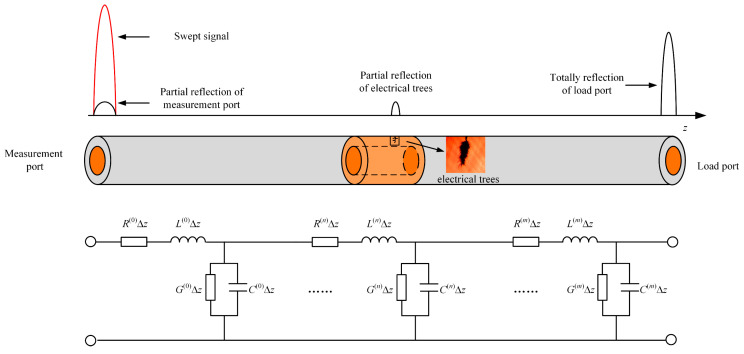
The principle of locating electrical trees by the BIS test.

**Figure 5 polymers-14-03785-f005:**
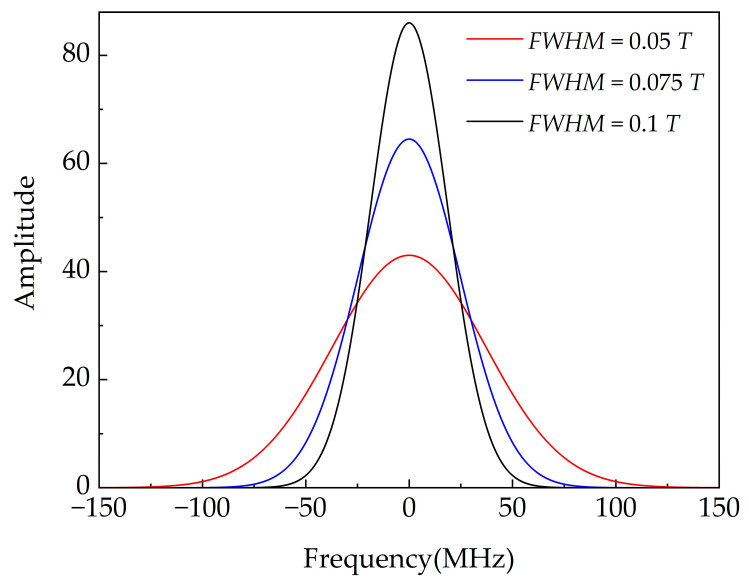
Amplitude–frequency diagram of Gaussian pulse with different FWHM.

**Figure 6 polymers-14-03785-f006:**
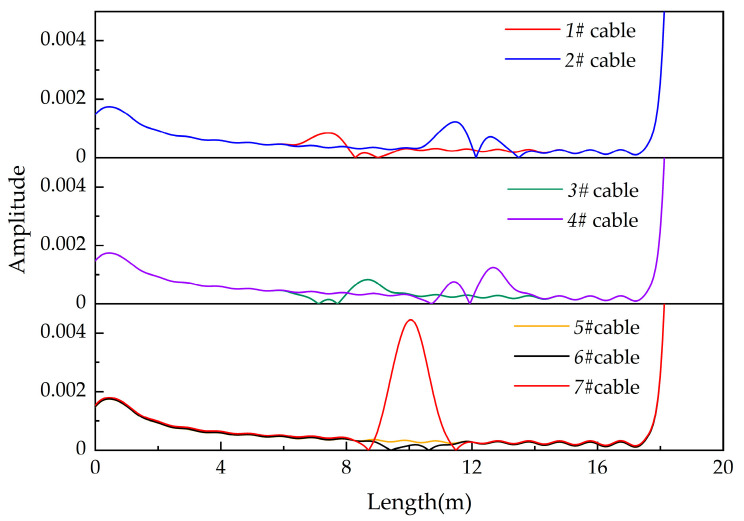
Location spectra of 1#–7# simulated cables.

**Figure 7 polymers-14-03785-f007:**
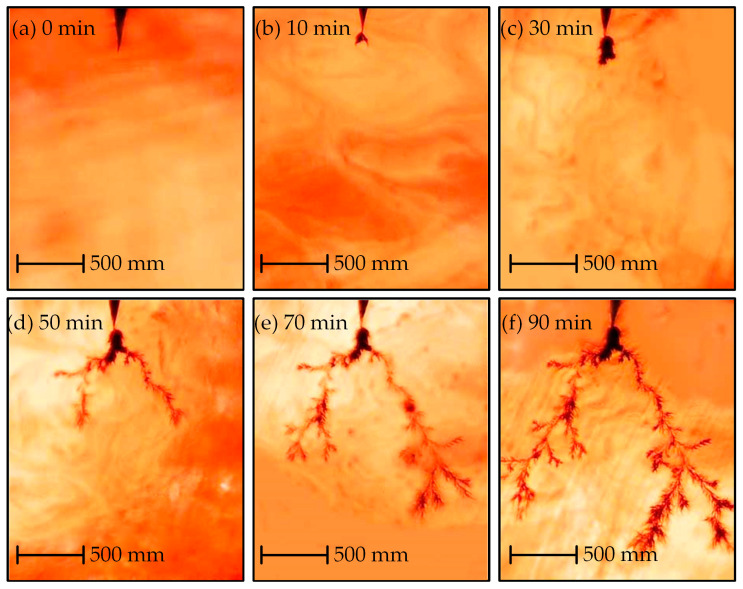
Growth characteristics of the electrical tree in sliced cable at 30 °C.

**Figure 8 polymers-14-03785-f008:**
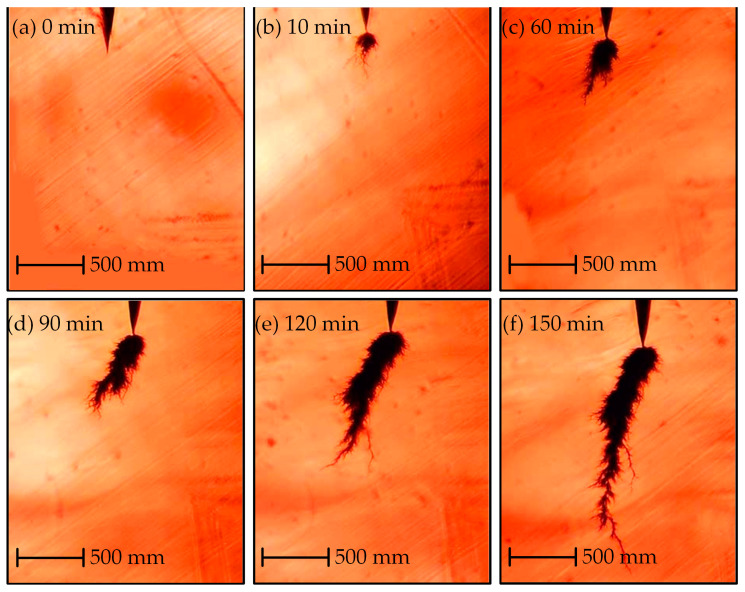
Growth characteristics of the electrical tree in sliced cable at 70 °C.

**Figure 9 polymers-14-03785-f009:**
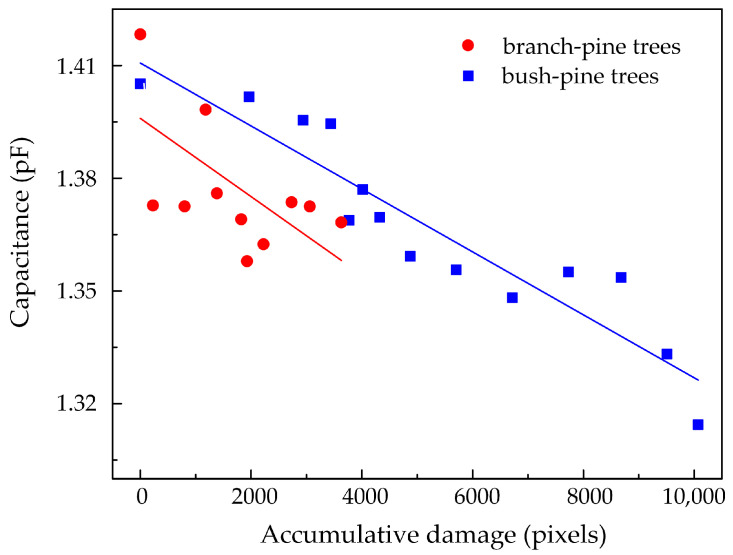
Capacitance at 50 MHz of 1 cm thick sliced samples during electrical tree growth.

**Figure 10 polymers-14-03785-f010:**
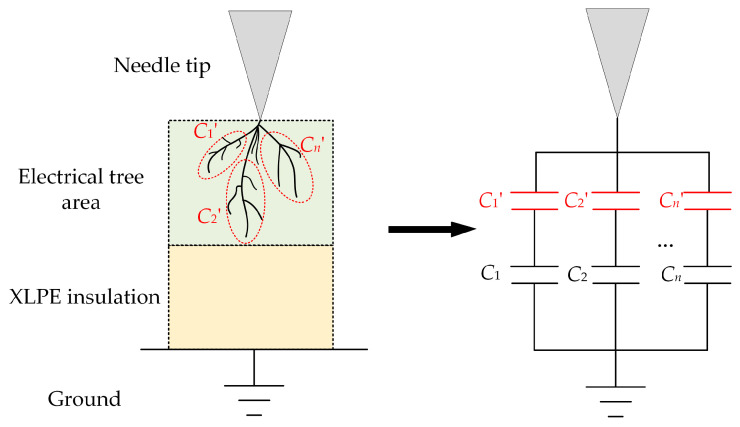
The model of distributed parameters during the growth of electrical trees.

**Figure 11 polymers-14-03785-f011:**
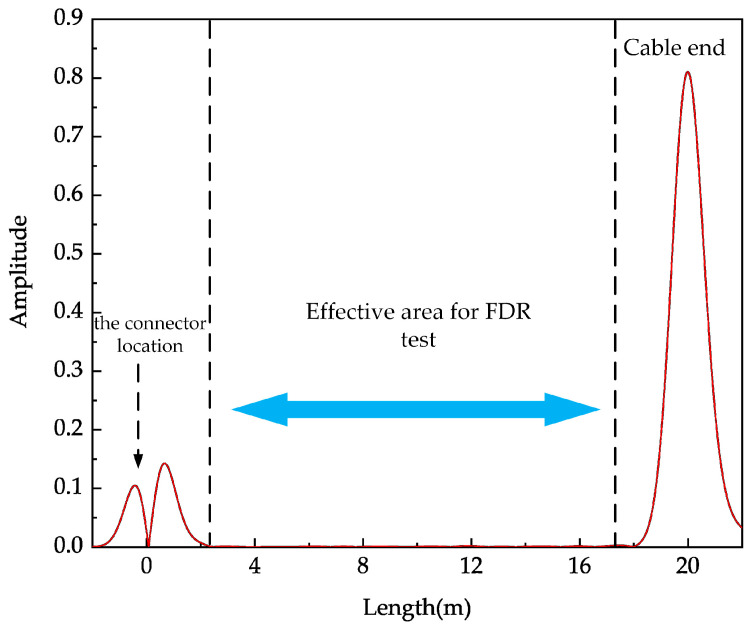
Location result of the tested cable.

**Figure 12 polymers-14-03785-f012:**
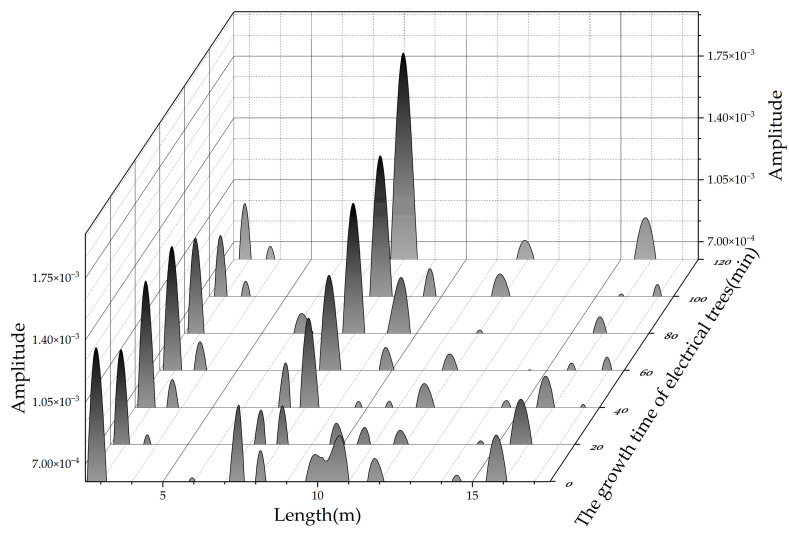
Location spectrum of cable with branch-pine trees at 8.2 m.

**Figure 13 polymers-14-03785-f013:**
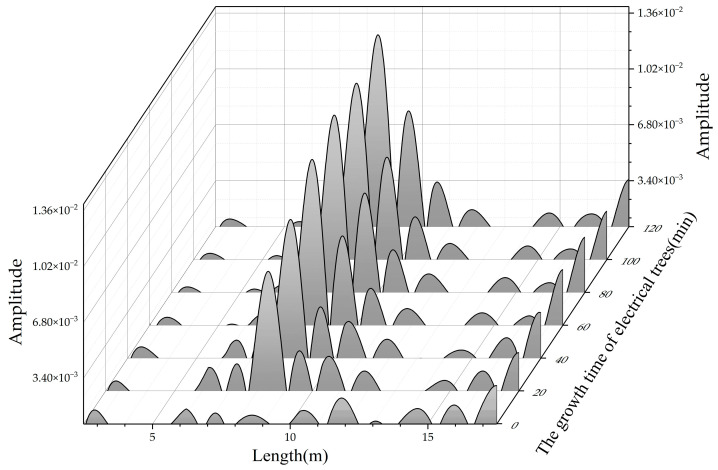
Location spectrum with bush-pine trees at 9.3 m.

**Figure 14 polymers-14-03785-f014:**
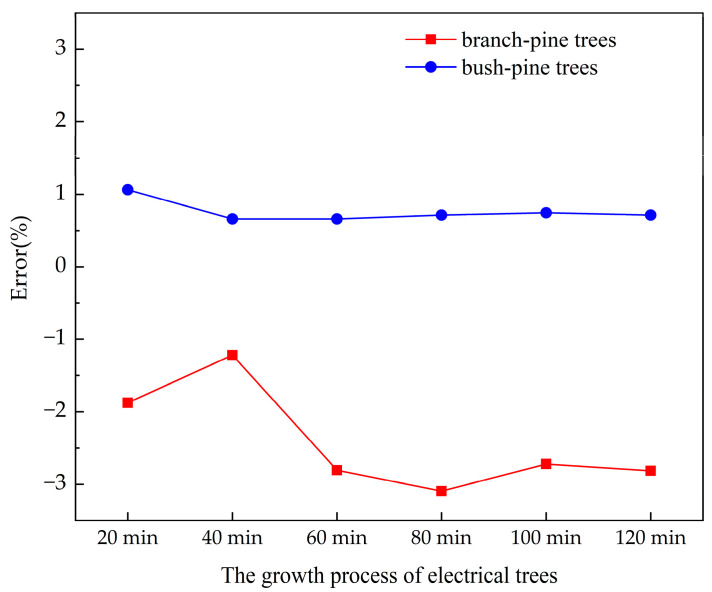
Location error of different electrical trees.

**Table 1 polymers-14-03785-t001:** The related parameters of the tested cable.

The Parameters of the Cable	Values
The radius of aluminum wire core *r*_c_ (mm)	4
The radius of copper shielding *r*_s_ (mm)	9
The relative dielectric constant of XLPE *ε*_r_	2.3
The conductivity of wire core *γ*_c_ (S/m)	3.538 × 10^7^
The conductivity of XLPE *γ*_r_ (S/m)	1 × 10^−17^
The conductivity of copper shielding *γ*_s_ (S/m)	5.714 × 10^7^

**Table 2 polymers-14-03785-t002:** The characteristics of 1#–7# simulated cable.

Number	Degradation Location (*m*)	Conductance (*S/m*)	Capacitance (*F/m*)
1#	7.98~8.02	*G* _0_	0.95 *C*_0_
2#	11.98~12.02	*G* _0_	0.90 *C*_0_
3#	7.98~8.02	*G* _0_	1.05 *C*_0_
4#	11.98~12.02	*G* _0_	1.10 *C*_0_
5#	9.98~10.02	1 × 10^12^ *G*_0_	*C* _0_
6#	9.98~10.02	1 × 10^13^ *G*_0_	*C* _0_
7#	9.98~10.02	1 × 10^14^ *G*_0_	*C* _0_

**Table 3 polymers-14-03785-t003:** The locating results of 1#–7# cables.

Number	Location of the Largest Peak (m)	Midpoint between Peaks (m)	Degradation Location (m)
1#	7.425	7.994	7.98~8.02
2#	11.484	12.041	11.98~12.02
3#	8.687	8.044	7.98~8.02
4#	12.647	12.041	11.98~12.02
5#	—	—	9.98~10.02
6#	10.197	—	9.98~10.02
7#	10.073	—	9.98~10.02

## Data Availability

The data presented in this study are available on request from the corresponding author. The data are not publicly available due to privacy reasons.

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
