# Peer review of "Locating Method for Electrical Tree Degradation in XLPE Cable Insulation Based on Broadband Impedance Spectrum"

_polymers, 2022, doi:10.3390/polym14183785_

Round 1
Reviewer 1 Report
1) Please check the denominator of equation (8).
2) Please discuss the data in table 2, and discuss more widely the data in the figure 6.
3) Discuss more widely the data in the figures 7 and 8.
4) Modify the word "capicitance" in row 394
5) Develop the "Conclusion" section.
6) It would be nice if you could insert a supplementary section, "Further works".
Reviewer 2 Report
Dear Authors,
The article in its current state can be interesting to the readers, provided that some changes & improvements are given. As a prelude, a scientific article is in particular a perspective of own results compared to results obtained by other teams.
The bibliography should be updated with recent articles of the field. Some references are given below :
1. Drissi-Habti M., Das RJ., Vijayaraghavan and Ech-Cheikh F., Numerical Simulation for Void Coalescence (Water Treeing) in XLPE Insulation of Submarine Composite Power Cables, Energies, 2020, vol. 13, issue 20, 1-17
2. DRISSI-HABTI M., S. Manepalli, On the way to the simulation for n-Voids Coalescence in XLPE Insulation of Submarine Hi-Voltage Smart Composite Power Cables, Energies MDPI Journal, 2020, Energies MDPI, 13(20):5472, DOI 10.3390/en13205472
Given that authors are carrying out tests on a phase (not on a cable), they must change "cable" by "phase" in all the figures and also in the text.
The results section is a collection of data with no comparison with what exists already in the bibliography. Therefore, what is asked in to systematically point out what is brought as innovations and how this compare with other teams research ... This will enhance the quality of the findings and emphasize up the guidelines of future research.
Reviewer 3 Report
The work presents a method of identifying the position of an electrical tree in the XLPE insulation of a cable. It is specified that the method would be useful in the case of large length cables. But it is not quite true that PD tests can harm insulation: nowadays there are methods that use reduced voltages and/or very low frequency (VLF) that do not affect insulation.
However, the experimental determinations were made on a cable of only 20 m. There is no question of the accuracy of the method in the case of the existence along the cable of several areas with electrical trees, and with different characteristics, which corresponds, in many cases to reality. The applied method gave good results in the laboratory, under the conditions of an induced defect and whose location is well known. The accuracy of this method must to be verify in the case of large length installed cables, because this is the goal of research.
The theoretical considerations regarding the modification of the parameters of a long line (conductance and capacitance) are correct, but the practical application of the method is questionable.
The work must be revised because a series of spelling errors appear and not only. So:
- In the Fig. 2 the high voltage terminal of divider is not connected; also in this picture, the word "Devider" must be replaced with "Divider";
- Instead alligator clips you can try to use a coaxial connection to the cable, taking in consideration the frequency level of measuring signal
- Replace “emersed” by “immersed” (row no. 106);
- At the beginning of subsection 3.1: the line with distributed parameters is applicable when the line/cable length is already greater than the wavelength of signal/10 (not when it exceeds the wavelength of signal);
- You must replace “distribution parameters” with “distributed parameters” (row no. 139, 141,147,155,167,172,227,238,299,315);
- The propagation constant k is not purely imaginary for lines with losses: the relation (2) expresses k not jk;
- You must note (in the relations which contain it) angular frequency with the appropriate Greek symbol, not with “w”;
- The expression in Eq. (5) is valid "in the case of impedance mismatching" but not "without..."
- You must correct the relation (8)! (see the denominator);
- You must write "the permeability and dielectric constant of vacuum (or of free space) but not "under vacuum"
- You must replace "diagiram" with "diagram" (row no. 212);
- add "magnitude" (in the row 259): "13 magnitude order";
-Replace "capicitance" with "capacitance" in the row no. 394.
However, it is recommended at least to use automatically spelling before sending the final version of the paper.
Round 2
Reviewer 2 Report
The paper is fine.